# Parallel Correlation Filters for Real-Time Visual Tracking

**DOI:** 10.3390/s19102362

**Published:** 2019-05-22

**Authors:** Yijin Yang, Yihong Zhang, Demin Li, Zhijie Wang

**Affiliations:** College of Information Science and Technology, Engineering Research Center of Digitized Textile & Fashion Technology, Ministry of Education, DongHua University, Shanghai 201620, China; 2171318@mail.dhu.edu.cn (Y.Y.); deminli@dhu.edu.cn (D.L.); wangzj@dhu.edu.cn (Z.W.)

**Keywords:** visual tracking, parallel correlation filters, reasonable distribution of correlation output, real-time

## Abstract

Correlation filter-based methods have recently performed remarkably well in terms of accuracy and speed in the visual object tracking research field. However, most existing correlation filter-based methods are not robust to significant appearance changes in the target, especially when the target undergoes deformation, illumination variation, and rotation. In this paper, a novel parallel correlation filters (PCF) framework is proposed for real-time visual object tracking. Firstly, the proposed method constructs two parallel correlation filters, one for tracking the appearance changes in the target, and the other for tracking the translation of the target. Secondly, through weighted merging the response maps of these two parallel correlation filters, the proposed method accurately locates the center position of the target. Finally, in the training stage, a new reasonable distribution of the correlation output is proposed to replace the original Gaussian distribution to train more accurate correlation filters, which can prevent the model from drifting to achieve excellent tracking performance. The extensive qualitative and quantitative experiments on the common object tracking benchmarks OTB-2013 and OTB-2015 have demonstrated that the proposed PCF tracker outperforms most of the state-of-the-art trackers and achieves a high real-time tracking performance.

## 1. Introduction

Visual tracking plays a core role in computer vision for its wide applications including video surveillance, robotics, driver-less vehicles, intelligent interaction, and various automatic systems [1]. The goal of visual tracking is to track the trajectory of the target that is initialized only by the bounding box from the first frame among the video sequences [2]. During tracking, the appearance of the target changes randomly and unpredictably when the target undergoes deformation, illumination variation, and rotation. It is one of the core issues determining the tracking accuracy and robustness in visual tracking. Although significant progress has been achieved in recent decades, visual tracking is still a challenging problem due to these factors.

In general, visual trackers can be broadly classified into two categories, generative trackers [3,4,5,6,7,8,9,10,11,12], and discriminative trackers [13,14,15,16,17,18,19,20,21,22,23,24,25]. Generative trackers describe the target in the real world by the target representation method in computer vision and establish a target appearance model dynamically to find a candidate most similar to the target appearance model in the video sequence. Therefore, generative trackers can reflect the similarity of the same kind of target [3]. However, it tends to produce a significant number of false positives and the learning process is complicated. Discriminative trackers extract the discriminative features of the target and utilize the method of classification in machine learning to search for the region most similar to the target and locate the position of the target [13]. Discriminative trackers can find the optimal classification surface between different categories and reflect the difference between heterogeneous data. The discriminative model is more elegant and efficient than the generative model. Therefore, the discriminative model-based methods have been widely applied in the visual tracking field.

Recently, the discriminative correlation filter-based (DCF-based) methods [13,14,15,16,17,18,19,20] ignited the interest of scholars for visual tracking due to their high accuracy and speed. The standard DCF methods utilize the only given sample by the initial bounding box and many synthetic samples to learn a linear classifier or linear regressor online to predict the position of the target [14]. These synthetic samples can be easily generated by the circular shift windows in the frequency domain. The DCF technique is a computationally efficient process in the frequency domain transformed by fast Fourier transform (FFT) [26,27]. Therefore, the DCF methods achieve significant improvement in terms of accuracy and speed. Despite their success, it is unreasonable that the DCF methods utilize a fixed learning rate to learn the linear classifier. The smaller learning rate means that the trained DCF cannot track the changes in the target appearance. The larger learning rate means that the trained DCF may produce model drifting caused by the significant changes in the target appearance, especially when the target undergoes deformation, illumination variation, and rotation. Furthermore, the correlation output of the DCF methods follows a 2-D Gaussian distribution which is smooth at the response peak. It is can easily cause the model to drift.

This paper proposes a novel parallel correlation filters (PCF) visual tracker to overcome these issues described above. The proposed method constructs two parallel correlation filters with different learning rates. The correlation filter with the bigger learning rate is used to track the appearance changes in the target. The correlation filter with the smaller learning rate is used to track the translation of the target. Through weighting the response maps of the two parallel correlation filters, the position and scale of the target can be predicted accurately. In addition, the standard DCF assumes that the correlation response obeys the Gaussian distribution which has an unreasonable smooth peak. The triangle distribution is also not suitable for the DCFs which has a small slope at the hillside of the correlation response. Hence, a new reasonable distribution of correlation response that organically merges the two distributions is proposed to replace the original Gaussian distribution to prevent the model from drifting. In this paper, the implementation codes are all open source in the following Github web: https://github.com/YijYang/PCF.git.

In summary, the main contributions of this paper include:A novel tracking framework named PCF that constructs two parallel correlation filters to simultaneously track the appearance and positional changes in the target;A new reasonable distribution of correlation output is proposed to replace the original Gaussian distribution to prevent the model from drifting to achieve higher tracking accuracy and robustness.Extensive qualitative and quantitative experiments on OTB-2013 [28] and OTB-2015 [29] benchmarks have demonstrated that the proposed PCF tracker performs better than most of the state-of-the-art trackers and simultaneously achieves a real-time tracking performance.

## 2. Related Works

Single visual object tracking is a quite common task in the field of computer vision. It has long been a focus area studied by many researchers. Plenty of visual trackers, including generative trackers, DCF-based trackers, and deep learning-based trackers, have been proposed in this field. In this section, the most representative and the most relevant works will be briefly discussed below.

### 2.1. Generative Trackers

The generative trackers utilize the features of the target extracted from the previous frame to dynamically learn the target’s appearance model, and then search for the region that best matches the prior appearance model among the candidate regions as the location of the target in the subsequent frame. Hence, the critical part to generative trackers is to search for the most similar candidate when the target has changed to some extent in a video frame. Up to now, many representative generative trackers [3,4,5,6,7,8,9,10,11,12] have been proposed, e.g., robust scale-adaptive mean-shift for tracking (ASMS) [3], incremental visual tracking (IVT) [4], fragment-based tracking (Frag) [5], multitask-based tracking (MTT) [6,7], distribution fields for tracking (DFT) [8] and distractor-aware tracker (DAT) [9]. ASMS is a real-time tracking algorithm implemented by minimizing the distance between two probability density histograms and it can track the scale of the target adaptively. However, it is easily distracted by similar objects in the surroundings. DFT builds a feature descriptor utilizing the distribution fields that smooth the objective function without breaking the information about the pixel value of the target. DAT trains the color histograms model of the target, distractors, and the background, and it uses the Bayesian function to compute the probability of each pixel belonging to the target, distractor, or background. However, these generative trackers do not take advantage of the background information and the performance of the generative models are limited by the dimensions of the target’s features. Therefore, the generative trackers could be easily distracted by background regions with similar appearances during tracking and show less discriminative power when the background is more complex.

### 2.2. DCF-Based Trackers

In contrast with generative trackers, DCF-based trackers effectively take advantage of the differences between the target and the background, and achieve better discriminative power and generative ability to separate the target from the background when the background is more complex. Recently, discriminative correlation filters (DCFs) have been widely applied in the visual tracking field in computer vision. Heads from signal processing, David S. Bolme and colleagues firstly introduced the correlation filter to the visual tracking fields [2]. They proposed the minimum output sum of squared error (MOSSE) tracker in the article ‘Visual Object Tracking using Adaptive Correlation Filters’. It produced significant performances, with tracking speed up to 700 frame rates. Thereafter, numerous improved algorithms [13,14,15,16,17,18,19,20] based on DCFs have been proposed with accurate and robust tracking performance by sacrificing the tracking speed, e.g., discriminative scale space tracking (DSST) [13], fast discriminative scale space tracking (FDSST) [13], spatially regularized discriminative correlation filters (SRDCF) [14], and efficient convolution operators with hand-crafted features (ECO-HC) [15] have demonstrated excellent performance on the popular benchmarks OTB-2013 and OTB-2015. DSST trains the translation and scale correlation filters using the handcrafted features extracted from an image sequence respectively. The improved FDSST utilized the principal component analysis (PCA) function to decrease the dimension of the extracted features to speed up the DSST. SRDCF adopts a spatial regularized component to penalize the correlation filters coefficients to solve the boundary effects. ECO-HC utilize a factorized convolution operator and a conservative update strategy to improve the speed and performance of the correlation filter. However, all these methods generate the training samples online and update the target appearance model with a fixed learning rate. Utilizing a fixed learning rate to update the model is unreasonable. DCF-based trackers cannot track the changes in the target appearance when the learning rate is too small. In contrast, trackers may produce model drifting when the target undergoes deformation, illumination variation, and rotation.

### 2.3. Deep Learning-Based Trackers

In several popular benchmarks [28,29], the deep convolutional neural networks (CNNs) have performed significantly well, which encouraged more recent works to either apply deep CNN features in the DCF algorithm [30,31,32,33,34] or design deep frameworks [35,36,37,38,39,40] for more accurate and robust visual tracking, e.g., learning multi-domain convolutional neural networks for visual tracking (MDNet) [33], multi-cue correlation filters for robust visual tracking (MCCT) [34], and convolutional residual learning for visual tracking (CREST) [35]. MDNet trains a multi-domain network composed of shared layers and multiple branches of domain-specific layers to track the target in different image sequences. MCCT constructs multiple-cue DCF experts to track the target independently achieving the strength of the different type of features. CREST reformulates DCFs as a one-layer convolutional neural network and exploits the residual learning method to achieve the benefit of end-to-end training and tracking. In contrast to hand-crafted features, learning DCF trackers that utilize deep CNN features remarkably increase their accuracy and robustness against the different degrees of change in the appearance model of the target. However, they suffer from high complexity and poor real-time performance. It is quite computationally expensive to extract deep CNN features with a high dimensionality or to implement deep architectures, resulting in poor real-time performance.

## 3. Proposed Approach

This section first provides a concise description of the standard discriminative correlation filter framework, then shows the implementation of the proposed PCF in detail. Finally, the algorithm framework of the PCF is described precisely in Algorithm 1.

### 3.1. Standard Discriminative Correlation Filter

The standard DCF [2] has been widely studied by many researchers due to its superior tracking accuracy and speed. The framework of the classical DCF is explicitly described in Figure 1. The DCF tracker trains a correlation filter model efficiently in the frequency domain by applying the machine-learning technique to distinguish the target from the background. It then updates the model online, exploiting the features extracted from the detected result in the current frame.

DCF learns the correlation filter $h_{t}^{l}$ online by exploiting n target samples $f_{1}^{l},f_{2}^{l},\ldots ,f_{n}^{l}$, which are efficiently generated by all cyclic shifts of the training sample. The objective function of the correlation filter $h_{t}^{l}$ is equal to an $L^{2}$ error function, which can be expressed as Equation (1):(1)$$\varepsilon (h_{t})=\sum_{k=1}^{n}{\left\|\sum_{l=1}^{d}h^{l}*f_{k}^{l}-g_{k}\right\|}^{2}+\lambda \sum_{l=1}^{d}{\left\|h^{l}\right\|}^{2}$$
where ∗ means the convolutional operator. k, l denote the number of training samples and the total dimension of the extracted features respectively. The expected correlation output $g_{k}$ is a 2-dimensional Gaussian distribution with the same size of $h_{t}^{l}$. The second term in this Equation is a regularization term which is utilized to prevent overfitting and λ(λ≥0) is a regularization parameter.

For computational efficiency, Equation (1) can be transformed into the frequency domain by fast Fourier transform (FFT). Then the objective function can be transformed as follows:(2)$$\underset{\left(H^{l}\right)}{\min}(\sum_{k=1}^{n}{\left\|\sum_{l=1}^{d}{\bar{H}}^{l}•F_{k}^{l}-G_{k}\right\|}^{2}+\lambda \sum_{l=1}^{d}{\left\|H^{l}\right\|}^{2})$$
where the capital letters denote the discrete Fourier transformations. $H_{t}^{l}$ denotes the correlation filter in the frequency domain and the overbar of ${\bar{H}}_{t}^{l}$ represents complex conjugation of $H_{t}^{l}$. The convolution operation ∗ can be efficiently implemented by element-wise product • in the frequency domain.

Therefore, through deriving Equation (2) and setting the derivative as zero, the final solutions are computed by Equation (5), which is implemented as follows:(3)$$A_{t}^{l}=\bar{G}F_{t}^{l}$$
(4)$$B_{t}=\sum_{k=1}^{d}{\bar{F}}_{t}^{k}F_{t}^{k}+\lambda$$
(5)$$H_{t}^{l}=\frac{A_{t}^{l}}{B_{t}},l=1,2,\ldots ,d$$
where t is the current step of the current video frame. In Equation (5), $H_{t}^{l}$ denotes the trained correlation filter in the frequency domain and l is the dimension value of the filter, here the dimension value is equal to the dimension of the extracted features. For computational efficiency, the solution can be divided into two part as Equations (3) and (4), $A_{t}^{l}$ represents the numerator of the filter, and $B_{t}$ represents the denominator of the filter. The overbar of $\bar{F}$ means the complex conjugation.

In order to adapt to the photometric and geometric variations of the target appearance, $A_{t}^{l}$, $B_{t}$ of the correlation filter is updated online by Equations (6) and (7) respectively:(6)$$A_{t}^{l}=(1-\eta_{0})A_{t-1}^{l}+\eta_{0}\bar{G}F_{t}^{l}$$
(7)$$B_{t}=(1-\eta_{0})B_{t-1}+\eta_{0}\sum_{k=1}^{d}{\bar{F}}_{t}^{k}F_{t}^{k}$$
where the scalar $\eta_{0}$ is a fixed learning rate.

For detecting the position of the target in the frame t, the features $Z^{l}$ are extracted from the region of the target pending detection. The responding correlation scores $y_{t}$ can be computed by Equation (8). Then the position of the maximum value of $y_{t}$ is regarded as the center position of the target area in the current frame.
(8)$$y_{t}=F^{-1}\left\{\frac{\sum_{l=1}^{d}{\bar{A}}_{t-1}^{l}Z^{l}}{B_{t-1}+\lambda }\right\}$$
where $F^{-1}$ denotes the inverse fast Fourier transform (IFFT).

In this paper, the DCF-based tracker ECO-HC [15] was chosen as the baseline tracker due to its excellent performance in terms of accuracy and robustness. The ECO-HC tracker has made several improvements on the basis of the standard DCF tracker. Firstly, it factorizes the convolution operation to reduce the model parameters. Secondly, it simplifies the generation of the training set by merging similar samples into a component to guarantee the diversity of samples. Thirdly, it employs the sparse update strategy to avoid the model drift problem where the update interval is set to 6. However, ECO-HC tracker has its own deficiencies. It is unreasonable to update the correlation model with a fixed learning rate when the sample appearance changes greatly due to deformation, illumination variation or rotation. Hence, ECO-HC tracker is difficult to track the target when the target appearance changes greatly. Furthermore, the Gaussian distribution with a smooth peak is utilized as the desired output of the correlation filter, which makes the result of the location inaccurate. In contrast to the ECO-HC, the proposed PCF tracker has been improved on this issue and obtained remarkable improvements on the benchmarks as shown in Figure 2.

### 3.2. Parallel Correlation Filters

The fundamental framework of the proposed PCF is explicitly described in Figure 3. In the initial frame, the PCF tracker trains two parallel correlation filters PCF1 and PCF2 online in a frequency domain utilizing the shared samples and the sharp correlation output. In the current frame, PCF1 and PCF2 are utilized to track the target respectively. Through weighting the response maps of PCF1 and PCF2, PCF tracker can detect the position of the target by applying the Newton method. Then it utilizes the Gaussian mixture model (GMM) [15] to generate a new sample set by adding a new sample or merging the two closest samples. As with the update strategy in the baseline tracker [15], it utilizes the new sample set to update the two parallel correlation filters with different learning rates every six frames.

PCF learns respectively two parallel correlation filters PCF1 $h1_{t}^{l}$ and PCF2 $h2_{t}^{l}$ by exploiting m target samples $f_{1k}^{l},f_{2k}^{l},\ldots f_{mk}^{l}$ with different weights $w_{kp}^{t}$ which are determined by the learning rate. The objective function of PCF1 can be expressed as Equation (9) and the objective function of PCF2 can be expressed as Equation (10):(9)$$\varepsilon (h1_{t})=\sum_{k=1}^{n}{\left\|\sum_{l=1}^{d}h1^{l}*\sum_{p=1}^{m}\left(w1_{kp}f_{kp}^{l}\right)-gs_{k}\right\|}^{2}+\lambda \sum_{l=1}^{d}{\left\|\omega h1^{l}\right\|}^{2}$$
(10)$$\varepsilon (h2_{t})=\sum_{k=1}^{n}{\left\|\sum_{l=1}^{d}h2^{l}*\sum_{p=1}^{m}\left(w2_{kp}f_{kp}^{l}\right)-gs_{k}\right\|}^{2}+\lambda \sum_{l=1}^{d}{\left\|\omega h2^{l}\right\|}^{2}$$
where k,p denote the number of samples in the training set and the circular shift set respectively. The sharp output $gs_{k}$ is merged by the Gaussian distribution and the triangle distribution which will be described in detail in Section 3.3. The second term of these two Equations is a regularized term and ω denotes the spatial regularization parameter.

Equations (9) and (10) can be transformed into the frequency domain efficiently by FFT. Due to the regularization parameter, ω breaks the closed solution of the objective functions, the solutions of $H1_{t-1}^{l}$ and $H2_{t-1}^{l}$ are obtained by the conjugation gradient (CG) iterative method [15]. Then for detecting the position of the target in the frame t, the shared features $Z_{t,pos}^{l}$ are extracted from the region of the target pending detection. The center position of the region is determined by the previously detected position. The corresponding correlation scores $y_{t,pos}$ can be computed by Equation (11). Then the position of the target in the current frame is optimized by the Newton iterative method.
(11)$$y_{t,pos}=F^{-1}\left\{\sum_{l=1}^{d}\left(\alpha H1_{t-1}^{l}Z_{t,pos}^{l}+(1-\alpha )H2_{t-1}^{l}Z_{t,pos}^{l}\right)\right\}$$
where *α* denotes the fusion factor. $H1_{t-1}^{l}$ and $H2_{t-1}^{l}$ represent two parallel correlation filters in the frequency domain.

After detection, the new sample is extracted from the tracking result. Then the GMM is utilized to compute the similarities between the new sample and the components of the training set. After that, a new training set is generated by adding the new sample or merging two closest samples. The weights $w_{kp}^{t}$ of the trained samples are updated by Equation (12).
(12)$$w_{kp}^{t}=(1-\eta )w_{kp}^{t-1}$$
where t denotes the *t*-th frame and η is the learning rate. Different learning rates mean different weights of samples.

For further detecting the scale of the target [13] in the frame t, the standard DCF is utilized to extract scale features $Z_{t,scale}^{l}$ with different scale factors and to compute the scale correlation scores $y_{t,scale}$ in Equation (13). Then the scale factor of the maximum value of $y_{t,scale}$ is utilized to compute the scale of the target in the current frame.
(13)$$y_{t,scale}=F^{-1}\left\{\frac{\sum_{l=1}^{d}{\bar{A}}_{t-1,scale}^{l}Z_{t,scale}^{l}}{B_{t-1,scale}+\lambda }\right\}$$
where $F^{-1}$ represents IFFT. ${\bar{A}}_{t-1,scale}^{l}$, $B_{t-1,scale}$ denote the numerator and the denominator of the scale correlation filter in the previous frame receptively.

### 3.3. Reasonable Distribution of the Correlation Response

The Gaussian distribution, the triangle distribution, and the merged distribution are described clearly in Figure 4a–c respectively. The desired output of the standard DCF methods follows a 2-D Gaussian distribution with a smooth peak. Due to the DCF training, the model using the synthetic samples generated by circular shift windows, the peak of the desired output should be sharp to avoid model drifting. The 2-D Gaussian probability density g(x,y) is expressed in Equation (14).
(14)$$g(x,y)=\frac{1}{2\pi \sigma^{2}}e^{-\frac{\left(x^{2}+y^{2}\right)}{2\sigma^{2}}},(-\frac{w}{2}\leq x\leq \frac{w}{2},-\frac{h}{2}\leq y\leq \frac{h}{2})$$
where σ represents the standard deviation and it is set to 1/16 in this paper. x, y denotes the coordinates for the figure pixels. w, h denotes the width and height of the figure, and $\left(\frac{w}{2},\frac{h}{2}\right)$ is regarded as the original point.

Different from the Gaussian distribution, the triangle distribution shown in Figure 4b has a sharp peak. However, it has a small slope at the hillside of the distribution. Due to the synthetic samples are generated by circular shift windows, the positions below the hillside of the distribution are regarded as the label of negative samples. The values of these positions should set nearly to zero. The 2-D triangle probability density s(x,y) is expressed in Equation (15).
(15)$$s(x,y)=(1-\frac{2}{w}|x|)(1-\frac{2}{h}|y|),(-\frac{w}{2}\leq x\leq \frac{w}{2},-\frac{h}{2}\leq y\leq \frac{h}{2})$$
where $\left(\frac{w}{2},\frac{h}{2}\right)$ is regarded as the original point.

Both Gaussian distribution and triangle distribution are not suitable for DCF. Hence, a new sharp distribution of the desired output is proposed to replace the original Gaussian distribution to achieve higher tracking accuracy and robustness. The 2-D merged probability density m(x,y) is expressed in Equation (16). The sharp distribution organically merges Gaussian distribution and triangle distribution by multiplication operators. It simultaneously combines the merits of these two distributions, with a sharp peak and a large slope at the hillside. Extensive experiments on benchmarks have demonstrated the effectiveness of the sharp distribution.
(16)$$m(x,y)=\frac{1}{2\pi \sigma^{2}}(1-\frac{2}{w}|x|)(1-\frac{2}{h}|y|)e^{-\frac{\left(x^{2}+y^{2}\right)}{2\sigma^{2}}},(-\frac{w}{2}\leq x\leq \frac{w}{2},-\frac{h}{2}\leq y\leq \frac{h}{2})$$
where $\left(\frac{w}{2},\frac{h}{2}\right)$ is regarded as the original point.

### 3.4. The Brief Outline of the PCF Tracking Algorithm

The proposed PCF tracking algorithm is briefly described in Algorithm 1. The PCF tracker first detects the position of the target, and then the scale correlation filter is utilized to refine the scale of the target. There are some differences between the scale and the position correlation filters. For the position filters, the PCF tracker extracts the position features from the first frame with a bounding box. It then uses the extracted features initializes two parallel correlation filters PCF1 $H1_{1}^{l}$ and PCF2 $H2_{1}^{l}$ online in the frequency domain with a sharp correlation output. From the second frame to the end frame of the sequence, PCF1 and PCF2 are utilized to track the target respectively. Through weighting the response maps of these two parallel correlation filters, the PCF tracker can detect the center position $P_{t}$ of the target by applying the NM. Then it utilizes the GMM to generate a new sample set. It also utilizes the features extracted from the new sample set to update the two parallel correlation filters $H1_{t}^{l}$, $H2_{t}^{l}$ with different learning rates $\eta_{t1}$, $\eta_{t2}$ every six frames. For the scale filter, the PCF tracker extracts the scale features from the first frame with different scale factors. It then uses the extracted features initializes the scale correlation filters $A_{1,scale},B_{1,scale}$. In the subsequent frame of the sequence, similar to the position filters, the scale filter utilizes the detected result $S_{t}$ to update the filter $A_{t,scale},B_{t,scale}$ by the learning rate $\eta_{0}$ every frame.


**Algorithm 1. PCF Tracking Algorithm**

**Input:**
 **1:** Image $I_{t}$. **2:** Detected target position $P_{t-1}$ and scale $S_{t-1}$ in the previous frame.
**Output:**
 **1:** Detected target position $P_{t}$ and scale $S_{t}$ in the current frame.
**Loop:**
 **1:** Initialize the PCF model $H1_{1}^{l}$, $H2_{1}^{l}$ and scale model $A_{1,scale}$,$B_{1,scale}$ in the first frame by Equations (9), (10) and (4), (5). **2: for**
$t\in \left[2,t_{f}\right]$
**do.** **3:  Position detection:** **4:**   Extract position features $Z_{t,pos}$ from $I_{t}$ at $P_{t-1}$ and $S_{t-1}$ by a search region. **5:**   Compute two parallel correlation scores $y1_{t,pos}$, $y2_{t,pos}$. **6:**   Merge the two correlation scores to $y_{t,pos}$ by Equation (11). **7:**   Set $P_{t}$ to the target position by Newton iterative method. **8:  Scale detection:** **9:**   Extract scale feature $Z_{t,scale}$ from $I_{t}$ at $P_{t-1}$ and $S_{t-1}$ by a search region. **10:**    Compute correlation scores $y_{t,scale}$ by Equation (13). **11:**    Set $S_{t}$ to the target scale that maximizes $y_{t,scale}$. **12:   Model update every six frames:** **13:**    Extract new sample features $F_{t,pos}$ and $F_{t,scale}$ from $I_{t}$ at $P_{t}$ and $S_{t}$. **14:**    Generate new training set by the GMM method. **15:**    Update the PCF model $H1_{t}^{l}$, $H2_{t}^{l}$ by the learning rate $\eta_{t1}$, $\eta_{t2}$. **16:**    Update the scale model $A_{t,scale}$, $B_{t,scale}$ by the learning rate $\eta_{0}$. **17:   Return**
$P_{t}$, $S_{t}$. **18: end for.**

## 4. Experiments

In this section, the implementation materials and parameter settings are described in detail. Comprehensive experiments are then performed on two benchmarks OTB-2013 [28] and OTB-2015 [29] to validate the effectiveness of the PCF tracker. Finally, the tracking failure cases of the proposed PCF tracker are analyzed briefly, and the future improvement works are further presented on this basis. The results of these experiments have demonstrated that the proposed PCF tracker performs better than most of the state-of-the-art methods.

### 4.1. Implementation Details

All experiments are performed on the same desktop (equipping with INTEL i5-4590 CPU with 8G RAM). The PCF tracker presented in this paper is implemented in MATLAB R2016a. The relevant parameters of the PCF tracker are briefly described in Table 1. For parameter adjusting, the extensive parameter setting experiments are conducted on OTB-2013 benchmark. Figure 5 shows that the PCF tracker achieved the best OPE accuracy at the point (0.9, 0.5). Therefore, the learning rate $\eta_{t2}$ of PCF2 is set to 0.5 in Equation (12), and the merging factor α of the two response maps is set to 0.9 in Equation (11). Following the baseline tracker [15], the learning rate $\eta_{t1}$ of PCF1 is set to 0.009 in Equation (12). For the generative model of the training set mentioned in Section 3.2, the initial weight $w_{kp}^{0}$ of the training sample is set to 1.0 and the number of the target samples m is set to 30. For the scale filter, the relevant parameters are set to the same values as the baseline tracker proposed in [19]. In order to make a fair comparison with the state-of-the-art trackers, the same parameters were used for all experiments on the benchmark data sets.

### 4.2. Ablation Experiments

To evaluate the effectiveness of progressively integrating the strategies proposed in this paper, the ablation experiments are conducted on OTB-2013 benchmark and the proposed PCF tracker is compared with the baseline tracker introduced in Section 3.1, the PCF with weighting fusion correlation response maps (PCF_WF tracker) proposed in Section 3.2 and the PCF with sharp output response (PCF_SR tracker) described in Section 3.3. The performance of these trackers is evaluated both on precision at different center location error thresholds and success rate at different overlap thresholds.

Figure 6 explicitly shows the comparison results on OTB-2013 benchmark in terms of the precision plots (PP) and success plots (SP) of one pass evaluation (OPE). The proposed strategies of weighting fusion and sharp output response in this paper either obtain significant improvement compared to the baseline tracker. In terms of the OPE of location error threshold (LET) at 20 pixels, the PCF_WR tracker and PCF_SR tracker achieve a gain of 3.3% and 3.1% compared to the baseline tracker respectively. From the aspect of the OPE of areas under the curve (AUC), the PCF_WR tracker and the PCF_SP tracker obtain a gain of 1.9% and 1.7% contrasted to the baseline tracker respectively. It indicates that both the two strategies are effective. Overall, the proposed PCF tracker integrating these two strategies performs the best performance in terms of precision and success rate. Concretely, the PCF tracker achieves 89.2%, 67.5% in the OPE of LET at 20 pixels and AUC, and compared to the baseline tracker, the PCF tracker achieves a remarkable gain of 4.3% and 3.2%.

Table 2 and Table 3 demonstrate separately the PP and SP of the OPE for the compared trackers in detail on eleven different attributes: scale variation (SV), illumination variation (IV), out-of-plane rotation (OPR), occlusion (OCC), background cluttered (BC), deformation (DEF), motion blur (MB), fast motion (FM), in-plane rotation (IPR), out-of-view (OV), and low resolution (LR). Obviously, the PCF_WF tracker and PCF_SR tracker all achieve various degrees of progress versus the baseline tracker, and perform well in real-time (average speed 44 FPS and 47 FPS respectively) in the benchmark. In Table 2, comparing PCF_WF tracker with PCF_SR tracker, the former performs better than the latter in terms of IV, OPR, BC, DEF, and IPR. However, from the aspects of SV, OCC, MB, FM, OV, and LR, the latter outperforms the former. This indicates that the strategy of weighting fusion is more robust to significant appearance changes in the target, especially when the target undergoes deformation, illumination variation, and rotation, while the scheme of sharp response is more accurate to the problem of model drifting caused by distractors or blur. In general, Table 2 and Table 3 indicate that the proposed PCF tracker acquires the best or the second results in terms of PP and SP of the OPE on all eleven attributes. Simultaneously, the PCF tracker runs an average speed of 42 FPS on all sequences of OTB-2013.

### 4.3. Experiments on OTB-2013

OTB-2013 is a common benchmark with 50 sequences that are divided into 11 different attributes: SV, IV, OPR, OCC, BC, DEF, MB, FM, IPR, OV, LR. On this challenge benchmark, the proposed PCF tracker is compared with 18 state-of-the-art trackers from the works: tracking-learning-detection (TLD) [2], distribution fields for tracking (DFT) [8], discriminative scale space tracking (DSST) [13], fast discriminative scale space tracking (FDSST) [13], spatially regularized discriminative correlation filters (SRDCF) [14], sum of template and pixel-wise learners (Staple) [16], compressive tracking (CT) [17], long-term correlation tracking (LCT) [18], locally orderless tracking (LOT) [19], least soft-threshold squares tracking (LSS) [21], visual tracking with online multiple instance learning (MIL) [22], scale adaptive kernel correlation filter tracker with feature integration (SAMF) [23], exploiting the circulant structure of tracking-by-detection with kernels (CSK) [24], high-speed tracking with kernelized correlation filters (KCF) [25], adaptive decontamination of the training set: A unified formulation for discriminative visual tracking (SRDCFdecon) [31], convolutional features for correlation filter-based visual tracking (DeepSRDCF) [32], Fully-convolutional Siamese networks for object tracking (SiamFC_3s) [36], object tracking via dual linear structured SVM and explicit feature map (DLSSVM) [37]. Only the ranks for the top 10 trackers are reported.

Figure 7 clearly illustrates the PP and SP of the OPE on three different attributes: DEF with 19 sequences, OPR with 39 sequences and IV with 25 sequences. As is shown in Figure 7, the proposed PCF achieves the best results among the top 10 trackers in these attributes. Specifically, the PCF tracker obtains 91.3% and 69.6%; 88.4% and 65.9%, and 82.1% and 62.3% on attributes DEF, OPR, and IV, respectively. It indicates that the PCF tracker is more accurate and robust to the significant changes in the target appearance compared with the other top nine trackers. Furthermore, the proposed PCF tracker is very effective for handling the challenges of deformation, illumination variation, and out-of-plane rotation.

The overall results on OTB-2013 benchmark among the top ten trackers are illustrated in Figure 8. Contrasted to the second tracker deepSRDCF-based on deep features, the proposed PCF tracker obtains the best ranks of 89.2%, 67.5% on the PP and SP of the OPE, and achieves a visibly gain of 3.9%, 7.5% in the PP and SP of the OPE respectively. Among the compared trackers employing handcrafted features, deep features, or combining these two features, the proposed PCF tracker achieves the best ranks and simultaneously runs a real-time speed of 41 FPS on a CPU. Furthermore, Table 4 and Table 5 explicitly shows the SP and PP of the OPE for the top 10 trackers on 11 challenge attributes respectively. As is demonstrated in Table 4 and Table 5, the PCF tracker outperforms the other top nine trackers on 10 out of 11 attributes of the OPE and obtains the best average precision (AP) and areas under the curve (AUC). It validates that the proposed sharp response strategy can effectively prevent the model from drifting and the proposed weighting fusion strategy is also very efficient to track the changes in the target appearance. Both strategies bring remarkable improvements in terms of accuracy and robustness. Besides, qualitative experiments are conducted on this benchmark and the results are reported in Figure 9. It further indicates that the proposed PCF tracker can accurately track the target with deformation, rotation, and illumination variation.

### 4.4. Experiments on OTB-2015

OTB-2015 is a more challenging benchmark than OTB-2013, including 100 videos with 11 different attributes: SV, IV, OPR, OCC, BC, DEF, MB, FM, IPR, OV, LR. The proposed PCF tracker is evaluated with 18 state-of-the-art trackers on this benchmark from the works: TLD [2], Incremental learning for robust visual tracking (IVT) [4], DFT [8], DSST [13], FDSST [13], SRDCF [14], Staple [16], CT [17], LCT [18], LOT [19], LSS [21], MIL [22], SAMF [23], CSK [24], KCF [25], SRDCFdecon [31], DeepSRDCF [32], and DLSSVM [37]. Only the ranks for the top 10 trackers are reported.

Figure 10 reports the PP and SP of the OPE determined by three challenges attributes: OPR with 63 videos, IV with 38 videos, and DEF with 44 videos. As is clearly illustrated in Figure 10, the proposed PCF provides the best results among the top 10 trackers in all three attributes. More particularly, the PCF tracker obtains 85.1% and 62.6%; 82.5% and 63.6%, and 83.0% and 62.5% on attributes OPR, IV, and DEF respectively. It again validates the effectiveness of the proposed PCF tracker for handling the issues of the significant changes in the target appearance.

Figure 11 shows the overall results for the top ten trackers on OTB-2015 benchmark. Among these compared trackers, the proposed PCF tracker obtains the best ranks in PP and SP of the OPE including the AP scores of 86.3% and the AUC scores of 64.7%. In addition, attributes-based evaluations are conducted on this benchmark. The results of this experiment are reported in Table 6 and Table 7. Specifically, the proposed PCF tracker achieves the top scores in terms of PP on 9 out of 11 attributes as shown in Table 7, and obtains the best ranks in terms of SP on all 11 attributes as demonstrated in Table 7. As is demonstrated explicitly in Table 6 and Table 7, the proposed PCF tracker obtains the best AP and AUC scores compared to the other top nine trackers, at the same time, the PCF tracker achieves a real-time performance running about 41 FPS on a CPU. It indicates the superior tracking performance for all 11 attributes. In general, the proposed tracker achieves a substantial improvement of the other top nine trackers in terms of accuracy, robustness and real-time performance. Furthermore, the qualitative experiments are also conducted in all videos of OTB-2015 benchmark. The results for the four representative videos are illustrated in Figure 12. Among the compared trackers, the proposed PCF tracker significantly outperforms the other top nine trackers in terms of location and scale estimation. It again validates that the effectiveness of the proposed PCF tracker when the target undergoes the situations of deformation, rotation, and illumination variation.

### 4.5. Analysis of the Failure Cases

Figure 13 shows the failure cases of the proposed PCF tracker. In the first row, the target underwent complete occlusions for long spans of time which caused the PCF tracker to drift off the target. While the LCT tracker can still track the target because of the re-detection strategy [18]. In the second row, the target underwent heavy occlusion with background clutters which also resulted in the PCF tracker failure to track the target. However, the trackers with deep features (e.g., deepSRDCF and SRDCFdecon) have high robustness in this situation. Furthermore, in the field of multi-object tracking, the top-down Bayesian formulation proposed in the work [41] can also solve the problems of occlusion and background clutters effectively. Hence, in the future works, the works [14,18,31,41] will be analyzed in detail and the strategies of these works will be merged into the proposed PCF tracker to address these issues.

## 5. Conclusions

In this paper, a novel PCF framework was proposed to address the issues of target appearance changes and model drifting. The proposed method constructed two parallel correlation filters with different learning rate. The correlation filter with the bigger learning rate was proposed to track the appearance changes in the target. The correlation filter with the smaller learning rate was applied to track the location of the target. Through weighting the response maps of the two parallel correlation filters, the position and scale of the target can be estimated accurately. Furthermore, a new reasonable distribution of correlation response that organically merges the Gaussian distribution and the triangle distribution was proposed to replace the original Gaussian distribution to prevent the model from drifting to achieve higher tracking accuracy and robustness. Extensive qualitative and quantitative evaluations on serval common benchmarks have demonstrated the competitive accuracy, robustness, and the superior tracking speed performance of the proposed PCF tracker compared to the state-of-the-art trackers. After analyzing the failure cases of the proposed PCF tracker, a new re-detection strategy will be studied in detail to improve the disadvantages of heavy or complete occlusions in visual tracking.

## Figures and Tables

**Figure 1 sensors-19-02362-f001:**
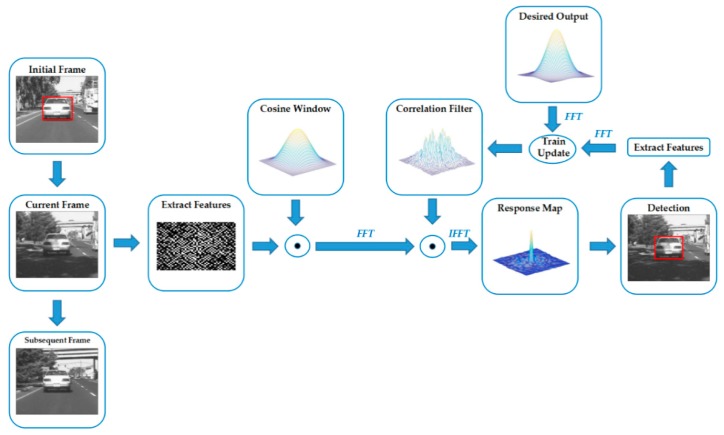
Illustration of the standard DCF framework. Firstly, the correlation filter is trained by the first frame with an initial bounding box and the desired output. Secondly, it extracts the features from the current frame and multiplies a cosine window to emphasis on the center region. The features are then transformed into the frequency domain by utilizing the fast Fourier transformation (FFT). Thirdly, the response map is obtained by multiplying the correlation filter and the extracted features. The response map is transformed into the time domain by applying the inverse fast Fourier transformation (IFFT). Finally, the position of the maximum value of the response map is regarded as the center position of the target in the current frame. The new features are then extracted from the detected result to train and update the correlation filter.

**Figure 2 sensors-19-02362-f002:**
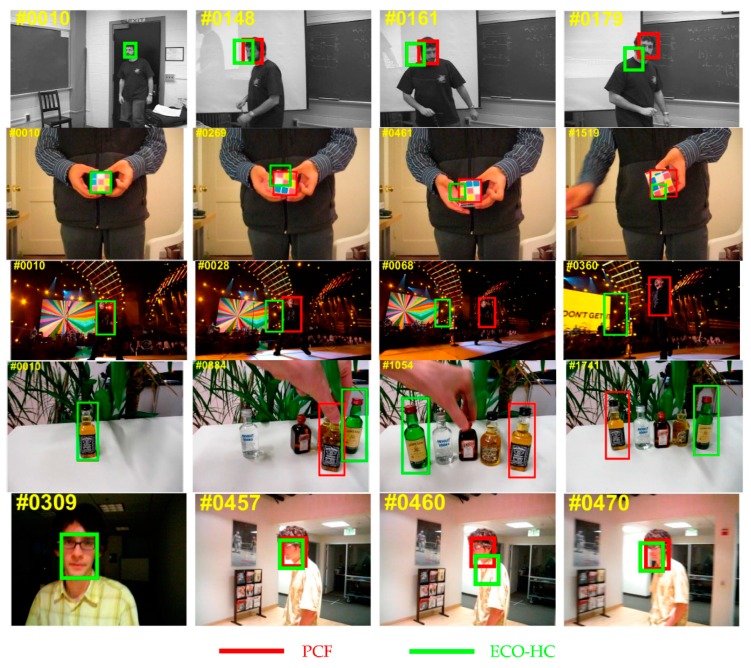
The qualitative comparisons of the proposed parallel correlation filters (PCF) tracker with the baseline tracker efficient convolution operators with hand-crafted features (ECO-HC) on five challenging sequences of the benchmarks. The results are marked in different colors. On all these five cases, PCF tracker performs better center position precision and overlap precision than the baseline tracker. PCF tracker successfully tracks the target when the target undergoes significant rotation, illumination variation, and deformation.

**Figure 3 sensors-19-02362-f003:**
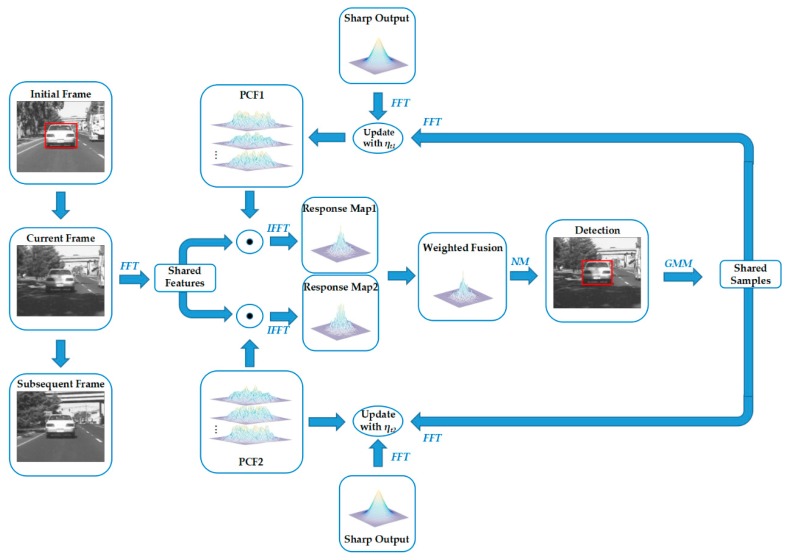
The flow chart of the proposed PCF framework. In the initial frame, the PCF tracker trains two parallel correlation filters PCF1 and PCF2 online in the frequency domain utilizing the shared samples and the sharp correlation output. In the current frame, PCF1 and PCF2 are utilized to track the target respectively. Through weighting the response maps of PCF1 and PCF2, the PCF tracker accurately detects the position of the target by applying the Newton method (NM). It then utilizes the Gaussian mixture model (GMM) to generate a new shared sample set by adding a new sample or merging the two closest samples. It utilizes the new sample set to update the two parallel correlation filters with different learning rates every six frames.

**Figure 4 sensors-19-02362-f004:**
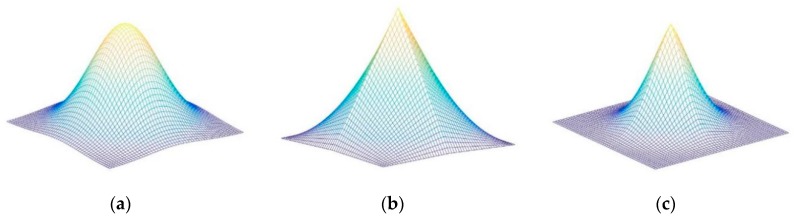
Different distributions of the desired correlation output: (**a**) 2-D Gaussian distribution; (**b**) 2-D triangle distribution; (**c**) 2-D merged distribution.

**Figure 5 sensors-19-02362-f005:**
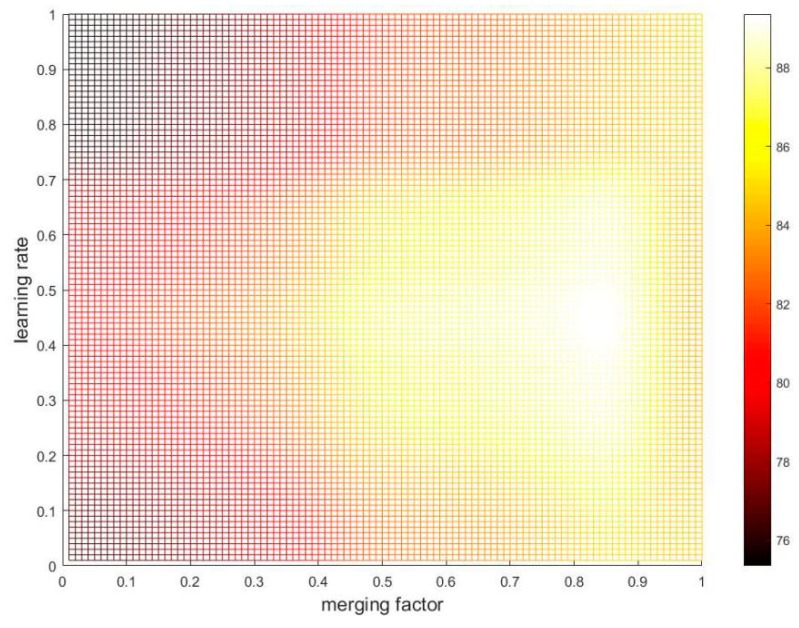
The results of the parameter setting experiments on OTB-2013 benchmark. The color bar denotes the one pass evaluation (OPE) accuracy percent (%) of location error threshold (LET) at 20 pixels. The horizontal and vertical coordinates represent merging factor α and learning rate $\eta_{t2}$ respectively. The points ranging from (0.1, 0.1), (0.1, 0.2), …, (1.0, 1.0) were obtained in experiments, others were obtained by the linear interpolation method. When the point $\left(\alpha ,\eta_{t2}\right)$ is set to (0.9, 0.5), the OPE accuracy is maximized.

**Figure 6 sensors-19-02362-f006:**
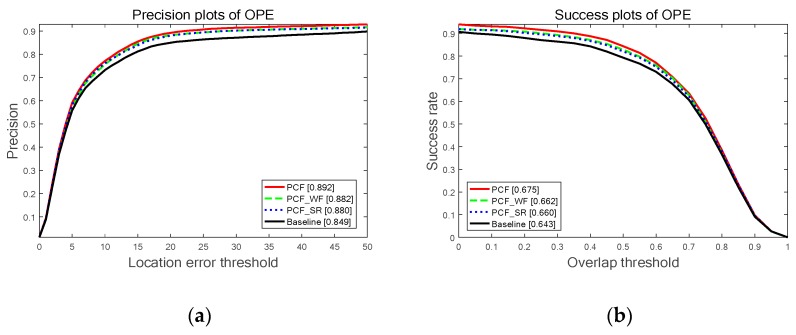
The results of the ablation experiments on OTB-2013 benchmark for the baseline tracker, the proposed PCF tracker and the trackers of progressively integrating the presented strategies: (**a**) The precision plots (PP) of one pass evaluation (OPE) and the values in brackets denote the OPE accuracy of location error threshold (LET) at 20 pixels; (**b**) The success plots (SP) of the OPE and the numbers in brackets are the areas under the curve (AUC).

**Figure 7 sensors-19-02362-f007:**
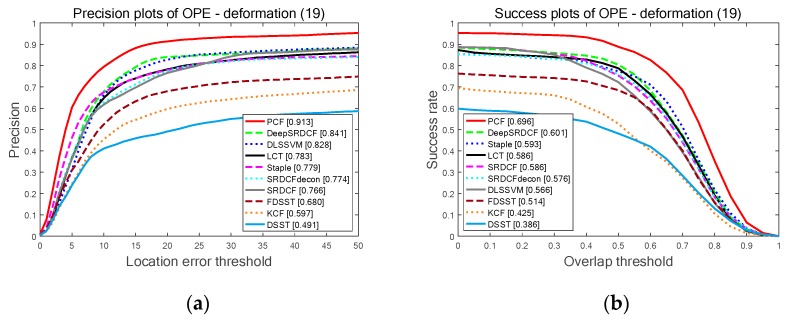
The quantitative results of the top ten trackers on OTB-2013 benchmark among three different contributions: DEF, OPR, and IV. Among the top ten trackers, the proposed PCF tracker obtains the best results on all three attributes: (**a**) The PP of the OPE on attribute DEF; (**b**) The SP of the OPE on attribute DEF; (**c**) The PP of the OPE on attribute OPR; (**d**) The SP of the OPE on attribute OPR; (**e**) The PP of OPE on attribute IV; (**f**) The SP of the OPE on attribute IV.

**Figure 8 sensors-19-02362-f008:**
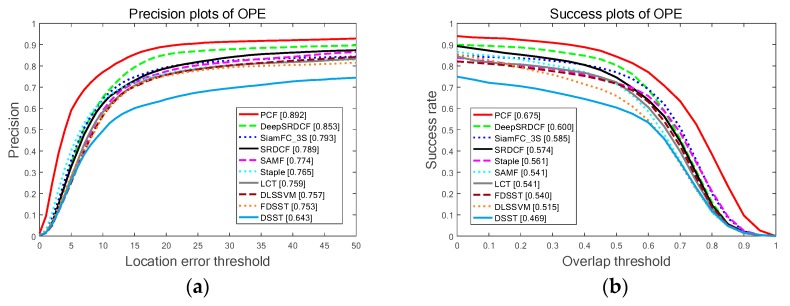
The comparing results for the top ten trackers with the whole 50 sequences on OTB-2013 benchmark: (**a**) The PP of the OPE on all sequences; (**b**) The SP of the OPE on all sequences.

**Figure 9 sensors-19-02362-f009:**
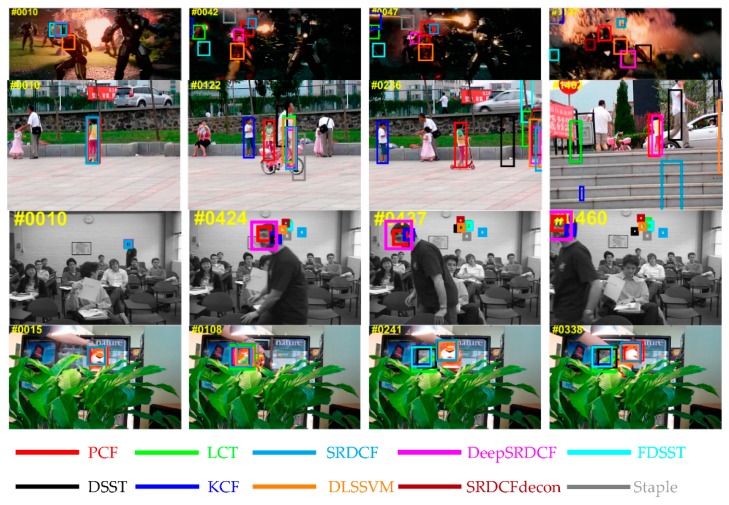
The qualitative experiments from four representative sequences on the OTB-2013 benchmark. The results for the top ten trackers are expressed in different colors. On these challenging sequences, the proposed PCF tracker achieves remarkable results compared to the other top nine trackers in terms of accuracy and robustness.

**Figure 10 sensors-19-02362-f010:**
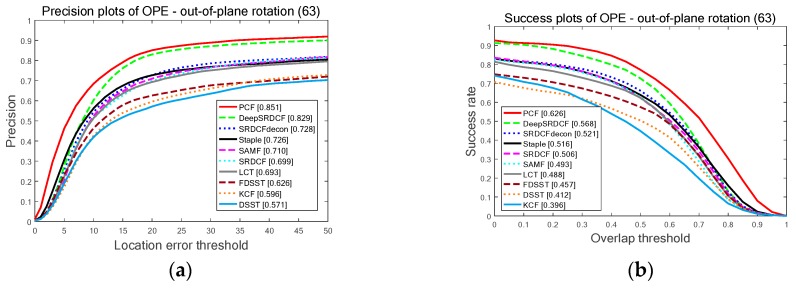
The quantitative results of the top ten trackers on OTB-2015 benchmark determined by three challenge contributions: DEF, OPR, and IV. Among the top ten trackers, the proposed PCF tracker achieves the top ranks in all three attributes: (**a**) The PP of the OPE on attribute DEF; (**b**) The SP of the OPE on attribute DEF; (**c**) The PP of the OPE on attribute OPR; (**d**) The SP of the OPE on attribute OPR; (**e**) The PP of the OPE on attribute IV; (**f**) The SP of the OPE on attribute IV.

**Figure 11 sensors-19-02362-f011:**
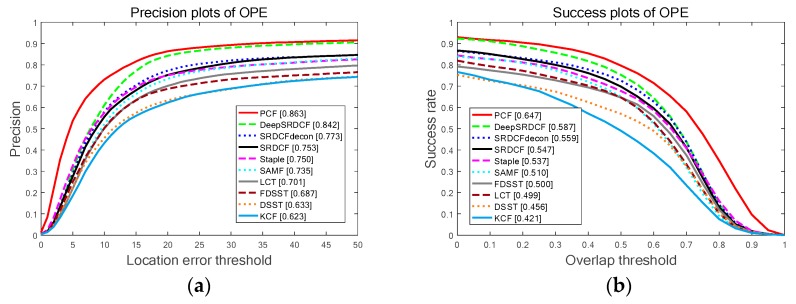
The comparing results for the top ten trackers with the entire 100 videos on the OTB-2015 benchmark: (**a**) The PP of the OPE on all videos; (**b**) The SP of the OPE on all videos.

**Figure 12 sensors-19-02362-f012:**
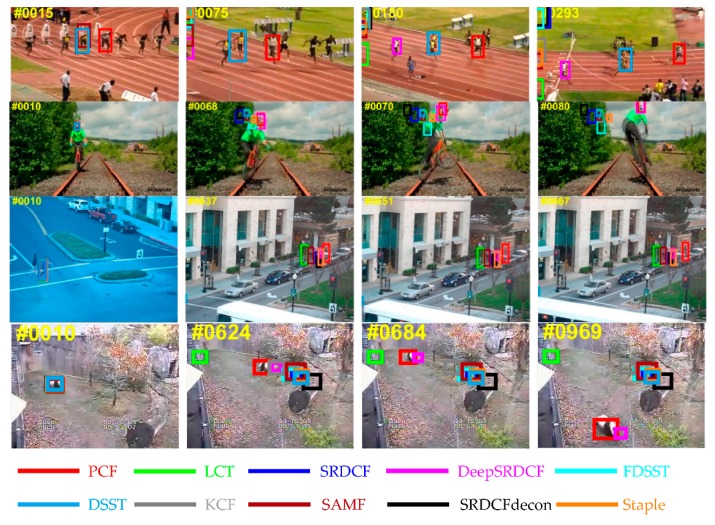
The qualitative experiments from four representative videos on the OTB-2015 benchmark. The results for the top ten trackers are marked in different colors. In these challenging videos, the proposed PCF tracker performs better than the other top nine trackers.

**Figure 13 sensors-19-02362-f013:**
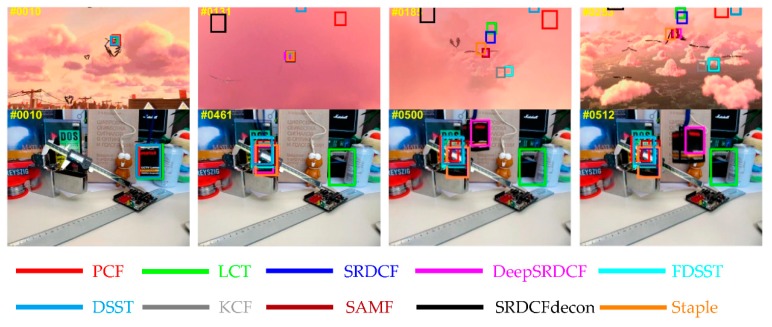
The failure cases from two representative sequences on the benchmarks. The tracking results for the top ten trackers are marked in different colors.

**Table 1 sensors-19-02362-t001:** The parameter settings of the proposed PCF tracker.

Parameters	$\eta_{t1}$	$\eta_{t2}$	α	$w_{kp}^{0}$	m
**Values**	0.009	0.5	0.9	1.0	30

**Table 2 sensors-19-02362-t002:** The PP of the OPE for the baseline tracker, the proposed PCF tracker and the trackers of progressively integrating the presented strategies on eleven different attributes (the ranks of results are expressed in %): scale variation (SV), illumination variation (IV), out-of-plane rotation (OPR), occlusion (OCC), background cluttered (BC), deformation (DEF), motion blur (MB), fast motion (FM), in-plane rotation (IPR), out-of-view (OV), and low resolution (LR). The last column is the average running speed (FPS) on all sequences of OTB-2013. The top three ranks are marked in **red**, **cyan**, and **blue** respectively.

Trackers	SV	IV	OPR	OCC	BC	DEF	MB	FM	IPR	OV	LR	Speed
Baseline	78.2	77.3	82.9	86.1	77.7	86.1	71.6	74.3	77.5	74.7	56.5	49
PCF_WF	80.3	80.9	87.2	89.1	82.4	91.2	73.9	74.3	81.3	79.7	71.0	44
PCF_SR	83.5	79.4	87.0	91.0	81.3	86.2	74.7	78.0	80.9	81.9	76.7	47
**PCF**	82.4	82.1	88.4	89.5	84.5	91.3	76.0	76.7	82.7	81.1	73.3	42

**Table 3 sensors-19-02362-t003:** The SP of the OPE for the baseline tracker, the proposed PCF tracker and the trackers of progressively integrating the presented strategies on different attributes (%): SV, IV, OPR, OCC, BC, DEF, MB, FM, IPR, OV, LR. The last column is the AUC which represents the average overlap accuracy of the OPE. The top three ranks are marked in **red**, **cyan**, and **blue** separately.

Trackers	SV	IV	OPR	OCC	BC	DEF	MB	FM	IPR	OV	LR	AUC
Baseline	60.3	59.5	61.6	64.4	58.5	65.1	57.7	57.8	58.1	60.1	35.8	64.3
PCF_WF	61.2	61.8	64.5	65.9	61.0	68.9	58.1	57.5	60.4	63.4	44.5	66.2
PCF_SR	62.8	60.4	64.2	67.4	60.3	65.6	58.3	59.0	59.6	64.7	48.1	66.0
**PCF**	63.2	63.1	65.9	66.9	63.5	69.6	60.1	59.3	62.2	64.5	48.8	67.5

**Table 4 sensors-19-02362-t004:** The PP of the OPE on OTB-2013 benchmark for the top ten trackers on eleven different attributes (%): SV, IV, OPR, OCC, BC, DEF, MB, FM, IPR, OV, LR. The last column is the average precision (AP) of LET at 20 pixels. The top three ranks are marked in **red**, **cyan**, and **blue**, respectively.

Trackers	SV	IV	OPR	OCC	BC	DEF	MB	FM	IPR	OV	LR	AP
**PCF**	63.2	63.1	65.9	66.9	63.5	69.6	60.1	59.3	62.2	64.5	48.8	67.5
DeepSRDCF	57.5	53.6	58.6	59.9	53.6	60.1	54.2	54.8	55.0	56.6	46.1	60.0
SiamFC_3s	57.0	50.5	55.9	56.1	54.2	53.5	49.0	51.5	55.2	54.5	52.9	58.5
SRDCF	54.1	51.5	55.1	57.4	53.0	58.6	51.6	51.4	51.7	51.1	41.0	57.4
Staple	53.7	54.3	55.7	55.9	54.5	59.3	47.8	45.2	53.5	40.5	36.2	56.1
SAMF	49.2	49.2	52.5	56.6	49.8	54.9	49.8	47.9	50.9	57.1	33.8	54.1
LCT	50.4	49.3	53.8	52.0	53.9	58.6	42.3	40.8	51.4	48.4	34.8	54.1
FDSST	52.1	51.7	52.1	50.7	57.3	51.4	48.8	47.5	52.3	47.9	28.8	54.0
DLSSVM	42.0	45.9	49.9	51.2	49.8	56.6	52.4	48.9	48.9	49.3	35.9	51.5
DSST	47.6	44.6	43.1	44.9	45.8	38.6	34.3	34.5	46.1	38.4	31.3	46.9

**Table 5 sensors-19-02362-t005:** The SP of the OPE on OTB-2013 benchmark for the top ten trackers on eleven different attributes (%): SV, IV, OPR, OCC, BC, DEF, MB, FM, IPR, OV, LR. The last column is the average overlap accuracy of the OPE. The top three ranks are marked in **red**, **cyan**, and **blue** respectively.

Trackers	SV	IV	OPR	OCC	BC	DEF	MB	FM	IPR	OV	LR	AUC
**PCF**	82.4	82.1	88.4	89.5	84.5	91.3	76.0	76.7	82.7	81.1	73.3	89.2
DeepSRDCF	80.7	76.5	84.6	85.9	76.7	84.1	74.9	75.6	80.0	78.2	74.9	85.3
SiamFC_3s	76.7	68.0	76.3	75.6	73.1	70.9	64.7	68.2	74.7	68.3	78.0	79.3
SRDCF	75.9	69.0	76.5	77.2	70.6	76.6	69.8	69.4	72.2	65.4	76.8	78.9
Staple	75.3	73.5	77.0	76.3	73.8	77.9	62.7	60.4	73.4	65.4	72.2	76.5
SAMF	73.4	69.0	75.9	80.7	65.6	74.0	64.8	63.0	72.9	66.8	70.8	77.4
LCT	72.1	68.6	76.0	71.1	73.5	78.3	55.9	52.0	72.4	56.7	68.1	75.9
FDSST	75.1	74.0	74.2	70.7	79.4	68.0	68.3	64.4	74.0	59.6	53.9	75.3
DLSSVM	67.4	66.5	75.6	72.3	69.6	82.8	72.9	66.8	74.2	61.9	82.6	75.7
DSST	68.2	60.0	60.0	60.1	59.8	49.1	43.2	42.1	65.1	48.6	69.3	64.3

**Table 6 sensors-19-02362-t006:** The PP of the OPE on the OTB-2015 benchmark for the top ten trackers on eleven different attributes (%): SV, IV, OPR, OCC, BC, DEF, MB, FM, IPR, OV, LR. The last column is the AP of LET at 20 pixels. The top three ranks are expressed in **red**, **cyan**, and **blue** respectively.

Trackers	SV	IV	OPR	OCC	BC	DEF	MB	FM	IPR	OV	LR	AP
**PCF**	83.9	82.5	85.1	84.4	82.8	83.0	82.3	82.4	80.7	79.1	89.2	86.3
DeepSRDCF	80.8	80.0	82.9	83.4	78.7	78.7	79.4	79.3	81.9	81.1	88.0	84.2
SRDCFdecon	75.6	76.2	72.8	71.5	74.2	72.4	74.2	72.1	71.3	59.6	78.2	77.3
SRDCF	71.8	73.4	69.9	66.7	68.0	68.2	70.0	71.1	71.7	55.0	79.5	75.3
Staple	73.2	76.4	72.6	69.9	71.4	71.0	64.9	67.7	75.7	59.2	77.3	75.0
SAMF	68.4	70.1	71.0	69.8	63.6	67.4	59.8	65.0	71.9	57.3	76.9	73.5
LCT	63.5	73.2	69.3	61.0	66.5	65.5	55.2	57.5	70.8	44.1	68.4	70.1
FDSST	64.1	70.9	62.6	59.9	74.5	57.1	63.8	65.6	69.2	52.5	65.3	68.7
DSST	61.8	62.1	57.1	53.7	62.2	49.7	51.4	51.9	64.1	50.1	68.8	63.3
KCF	58.0	63.1	59.6	52.6	62.3	55.9	50.6	54.9	63.5	37.2	67.1	62.3

**Table 7 sensors-19-02362-t007:** SP of the OPE on OTB-2015 benchmark for the top ten trackers on eleven different attributes (%): SV, IV, OPR, OCC, BC, DEF, MB, FM, IPR, OV, LR. The last column is the average precision of LET at 20 pixels. The last column is the AUC of the OPE. The top three ranks are expressed in **red**, **cyan**, and **blue** respectively.

Trackers	SV	IV	OPR	OCC	BC	DEF	MB	FM	IPR	OV	LR	AUC
**PCF**	61.6	63.6	62.6	61.6	63.0	62.5	63.2	61.4	59.3	56.2	56.0	64.7
DeepSRDCF	56.5	57.0	56.8	57.5	55.7	53.6	58.9	57.8	54.6	54.1	50.5	58.7
SRDCFdecon	54.3	55.7	52.1	52.7	55.0	52.2	57.2	54.5	49.9	48.0	46.7	55.9
SRDCF	51.8	54.2	50.6	50.7	51.4	50.4	52.9	54.4	50.0	42.8	46.8	54.7
Staple	51.3	55.4	51.6	51.3	52.3	52.3	50.4	50.7	51.4	45.0	41.1	53.7
SAMF	46.0	49.2	49.3	48.9	47.5	46.9	47.3	48.9	48.9	44.8	39.3	51.0
FDSST	46.4	49.6	45.7	44.3	54.2	42.8	47.6	49.6	49.6	41.5	39.1	50.0
LCT	44.7	50.0	48.8	45.0	49.5	46.7	44.2	45.7	48.3	37.8	37.1	49.9
DSST	43.3	45.1	41.2	40.4	46.1	37.2	41.0	41.6	44.7	39.0	33.8	45.6
KCF	36.4	41.0	39.6	36.8	43.4	39.3	38.2	40.4	42.2	31.8	29.0	42.1

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
