# Peer review of "Parallel Correlation Filters for Real-Time Visual Tracking"

_sensors, 2019, doi:10.3390/s19102362_

Reviewer 1 Report

You have done a good job with interesting results. However, there are some points you should improve before your publication gets accepted:

- Improve language: You have made many mistakes using the English grammar and spelling of certain words. I've marked some of them in the attached file. Consider to hand your text to a native speaker in order to improve the overall language. 

- Consider not using red and green color within the same figure / table. This is quite tough to distinguish by people suffering from dyschromatopsia.

- The plot and explanation on page 8 of 20 ist very confusing. Consider to refactor this part. You are talking about distributions, but you are presenting a probablity density function. Moreover, it is not clear what the second subfigure (b) shows.

- Be clear in your figure captions. A presented figure should be understandable without having to read a certain paragraph.

Author Response

Thank you so much for your positive and constructive comments and suggestions. We provide a point-by-point response to your comments in the attached file. 

Reviewer 2 Report

The paper is in general well written and has good theoretical support with convincing results.

The subject is relevant for the readers and the proposed approach is sound with convincing results. I have no major concern in relation to the text and the research itself. 

A have just a couple of very minor suggestions for enhancements:

1)      Please, refrain from using the first person, “we, our, us”, please prefer using third person or passive voice instead (direct speech is always better, but never with first person!).

2) Algorithm 1 in section 3.4 could be better and more detailed explained.

Author Response

(The authors gave the same response as above.)

Reviewer 3 Report

In general, paper is updated reasonably well, however, there are still typos and editing issues in the manuscript. Additionally, there are few comments on parameter settings and relevant reference listing for quality of the paper.

In the line 234, page 7, why did you choose different learning rate every 'six' frames? What is the reason to choose the learning rate of \eta_1 and \eta_2, the merging factor and how these values affect the performance? There must be discussion for the table 1 how to set up these values.

There are typos and grammatical errors in the manuscript. For instance, the upper letter is used for "where" after the equation, see the line 294, 302. Please check typos and grammar errors again.

It is good to add some of recent relevant application of KCF papers in the introduction such as extension to the multi-object tracking [*].

[*] D. Y. Kim, B.-N. Vo, and B.-T. Vo, "A labeled random finite set online multi-object tracker for video data," Pattern Recognition, vol. 90, pp. 377-389, 2019.

Author Response

(The authors gave the same response as above.)
